# Persons Experiencing Homelessness during Extreme Temperatures: Lessons for Promoting Socially Inclusive Adaptive Capacity

**DOI:** 10.3390/ijerph21080984

**Published:** 2024-07-27

**Authors:** Courtney Cronley, Amanda Fackler, Jennifer M. First, Sangwon Lee, Iris Tsouris

**Affiliations:** 1College of Social Work, University of Tennessee, Knoxville, TN 37996, USA; afackler@vols.utk.edu (A.F.); slee192@vols.utk.edu (S.L.); 2School of Social Work, University of Missouri, Columbia, MO 65201, USA; firstj@missouri.edu; 3Yale University, New Haven, CT 06520, USA; iris.tsouris@yale.edu

**Keywords:** homelessness, green infrastructure, extreme temperature, urban planning, adaptive capacity

## Abstract

Climate change and increasing extreme temperatures present unique challenges to persons experiencing homelessness (PEH), including heightened physical and psychological harm. While green and urban infrastructure has emerged as one possible mitigation strategy, homeless populations are rarely included in municipal disaster planning or infrastructure research. This study used in-depth interviews with PEH (*N* = 42) during the summers of 2022 and 2023. Questions were designed around phenomenological methods to explore the individuals’ firsthand descriptions of the lived experience of coping during extreme temperatures within a mid-size city in the Southeastern United States. Our findings highlight how social exclusion within the built environment reduces PEH’s adaptive capacity and increases the physical and psychological risks of extreme temperatures, namely through limiting and policing scarce resources and restricting the mobility of PEH. In contrast, public transit provided relief from extreme temperatures. Implications from our findings include the need for attention on inclusive green urban infrastructure, including increased placement and access to shade, public water, mixed-use daytime sheltering models, and the installation of lockers to increase capacity to maintain supplies and gear necessary for enduring extreme temperatures. Findings also highlight the challenges of designing inclusive green infrastructure and the importance of de-stigmatizing homelessness and building more housing and income support to increase adaptive capacity for an entire community in the context of a rapidly warming climate.

## 1. Introduction

Homelessness continues to be a significant societal and public health issue in the United States (US). According to the 2023 Annual Homelessness Assessment Report (AHAR) to Congress, approximately 653,000 persons are experiencing homelessness (PEH) on any single night in the US, a 12% increase since 2022 [1]. Over half of those individuals (59%) were experiencing homelessness in an urban area, and 39.9% were unsheltered, meaning their primary nighttime location was a place not designated for habitation, such as in a car or on the street [1]. Almost half of all unsheltered homeless individuals were aged 35–54 and were, on average, older than their sheltered counterparts [1]. Approximately 143,000, or 31%, of all PEH in 2023 reported experiencing chronic patterns of homelessness—the highest number since 2007 [1]. Among those experiencing chronic homelessness, two-thirds were unsheltered.

As populations of PEH have risen, so, too, have climate-induced weather risks, such as extreme temperatures and severe storms, producing excessive precipitation, hail, wind hazards, and flooding. PEH face some of the highest risks for negative impacts of climate change and extreme weather conditions due to chronic exposure from living in urban and outdoor environments, combined with stigma and policing of public spaces and social isolation [2]. Nevertheless, there is a shortage of theoretically framed studies asking PEH directly about the impact of extreme weather, with a specific focus on how the urban environment influences their ability to cope. This study, informed by the dual frameworks of social exclusion [3] and adaptive capacity [4], addresses this literature gap and utilizes qualitative methods of interpretive phenomenology to answer the following question: What are the lived experiences of persons experiencing homelessness during extreme weather conditions within one Southeastern US city?

### 1.1. Social Exclusion and Adaptive Capacity

The social and economic factors underlying homelessness make PEH uniquely vulnerable to health risks related to climate-change-related extreme weather. Homelessness has been described as a system of social exclusion or the systematic marginalization of specific sub-groups of the population such that they are denied access to resources and or discouraged from participating in the larger community [5]. Social exclusion originated in poverty scholarship to explain the lack of upward economic mobility driven by the denial of basic human and citizen rights, particularly for communities of color and persons who are very low-income [6]. Shinn [7] argues that poverty and social exclusion are the key causes of homelessness in the US, where income inequality and racial disparities in wealth are higher than in other developed nations and where there are fewer and less comprehensive social welfare programs. Individuals with less income, racial minorities [1], or those who have experienced adverse life events such as childhood homelessness, foster care, criminal justice involvement, or incarceration [8] are overrepresented in US homeless populations. Without a federal right to housing, PEH face waitlists for housing and sleep in emergency shelters, cars, on the couches of family or friends, or outdoors. Those sleeping in outdoor locations and who have been homeless for longer periods of time show more fractured social support networks [9].

The process of social exclusion may also explain higher rates of poor health in PEH [10], suggesting that lower-quality social interactions and the challenges of living on the street influence their health behaviors and contribute to more health risk behaviors. For example, PEH may avoid formal health care after facing stigma related to their housing situation or face transportation barriers that limit their access to resources [11]. The same factors apply in the context of climate-induced weather risks and lead to reduced adaptive capacity among PEH [12]. Adaptive capacity is an increasingly important construct in our understanding of climate change and can be used to explain differential risks and impacts of climate change across societies [13]. Adaptive capacity is defined as social systems’ abilities to adapt to change in response to a changing climate [13]. Resources that are cited as increasing adaptive capacity, such as alternative housing, transportation, and information [13], are all resources that individuals experiencing social exclusion and PEH lack. As Thomas and colleagues [12] argue, access to resources is predicated on both availability as well as social positionality and power. When individuals face systemic marginalization due to positionalities of race, class, or gender, they may be unable to access the available resources. The result is less adaptive capacity and greater vulnerability in the context of climate change. Research considering social exclusion, marginalization, and climate change vulnerability has been conducted largely in developing countries (see [14] for example), and the idea of reduced adaptive capacity based on social exclusion among PEH, particularly within the US, remains a relatively under-explored phenomenon.

### 1.2. Climate Change, Homelessness, and Health 

The need to focus on adaptive capacity for PEH is growing increasingly critical. The impact of human activities on climate is expected to cause continued significant global changes, including more frequent and intense extreme weather events, rising sea levels, and changes in precipitation patterns [15]. These changes are likely to result in increased temperatures and extreme weather-related fatalities and injuries, increases in and exacerbation of mental health conditions, and an increase in illnesses related to a lack of food and clean drinking water [15]. Urban areas are particularly vulnerable to the effects of climate change, such as longer and more frequent heat waves, increased air pollution, and damage to critical infrastructure [16]. These extreme weather events can have devasting effects on vulnerable populations in urban areas and pose significant challenges for individuals experiencing homelessness, who already face numerous barriers to accessing shelter, safety, and basic resources [17].

PEH are more likely to suffer from chronic health conditions and compromised health, making them especially vulnerable to the effects of severe weather and increasing their risks of infections and diseases [17]. Unsheltered homeless communities are particularly less likely to access adequate shelter and protection from extreme heat, cold, and storms [18]. The impacts of extreme weather events on the unsheltered homeless communities extend beyond immediate physical safety and health concerns. Prior research has shown that they are often displaced, with their few belongings being destroyed by flooding or wildfires, further exacerbating their vulnerability and hindering their ability to escape the cycle of homelessness [18].

The stress of choosing between necessities, such as food, water, housing, and energy, can profoundly affect both adults’ and children’s mental health and well-being [19]. Even more so, environmental factors can influence access to resources like food, water, and shelter. Lack of access can profoundly affect the mental health and well-being of homeless individuals [2]. These findings underscore the complex interplay between climate-induced weather events, social networks, and access to resources and shelter in shaping the experiences of unsheltered homelessness and emphasize the need for holistic approaches to support this vulnerable population [18]. 

### 1.3. Temperature Extremes 

Previous studies on weather events and homelessness have focused on the impact of temperature extremes on the health and well-being of homeless individuals and have shown that PEH are particularly vulnerable to health risks associated with temperature extremes [2,20,21,22]. Specifically, low temperatures increase the risk of hypothermia and frostbite among PEH [23] while high temperatures increase dehydration and heat exhaustion rates [24,25]. 

The heat island effect is prevalent in urban and suburban built environments found to retain heat, thereby re-emitting heat and causing temperatures to rise by 5–11 degrees Celsius in urban regions [26]. This, in turn, leads to a shortage of cool spaces, which is particularly challenging for communities residing in areas with a higher concentration of poverty and racial minorities, where tree canopy coverage is typically lower [26,27]. Prolonged exposure to elevated temperatures can exacerbate pre-existing health conditions, leading to heatstroke, dehydration, and a range of issues impacting the cardiovascular, renal, and respiratory systems [25]. PEH may be particularly vulnerable to heat-related illnesses because they have limited adaptive and coping resources, such as access to indoor air conditioning spaces [28]. 

In colder weather, the risk of hypothermia and respiratory and cardiovascular issues increases, which in turn leads to higher mortality rates and increased prevalence of hypothermia, respiratory, and cardiovascular diseases [29]. A study conducted in Nashville, TN, in the Southeastern US, found that self-reported general health and well-being were the lowest during winter months [30].

### 1.4. Changes in Precipitation and Water Access 

The homeless community is vulnerable to various risks associated with flooding and heavy rain, including infectious disease outbreaks, drownings, and a heightened incidence of anxiety and depression [31]. Homeless individuals residing in encampments near rivers or bodies of water face considerable threats and hazards to their person and personal property [32]. They must also contend with the challenges of managing wet gear, having dry and appropriate footwear, and the risk of getting trapped in floodwaters [2,32]. 

The lack of access to drinking water is also a significant risk factor during extreme heat [25,33]. Stigma, social exclusion, and discrimination can impede water access for PEH [8]. Water insecurity during heat waves has been documented to result in a plethora of physical and mental health ailments, ranging from dehydration, sunburn, heat exhaustion, and heat cramps to more severe heat stroke [33]. Moreover, prolonged exposure to such conditions has been found to significantly impact the overall health status of individuals, leading to the onset of cardiovascular diseases, high blood pressure, diabetes, and hypertension [25,33]. 

### 1.5. Green and Urban Infrastructure 

Green and urban infrastructure refers to using natural elements, such as trees and green spaces, and built infrastructure, such as permeable pavement and green roofs, to enhance urban environments and promote sustainable living [34]. These strategies aim to improve urban residents’ quality of life, reduce urbanization’s negative impacts on the environment, and promote community-level resilience to climate change [34,35]. In the context of homelessness, green and urban infrastructure can offer solutions to mitigate the risks of extreme weather events and provide access to essential resources and services for vulnerable populations [27,36]. However, climate change and extreme weather events can damage or destroy green and urban infrastructure, such as parks, green spaces, community centers, and affordable housing units [27].

The effectiveness of green and urban infrastructure in mitigating the risks associated with extreme weather conditions for PEH is a growing area of research. Some studies have shown that green and urban infrastructure can provide cooling and shelter during heat waves. However, others have questioned the accessibility and practicality of these solutions [37,38]. Some of the accessibility and practicality issues with green infrastructure include the availability of land, cost, maintenance, and the time required for the infrastructure to mature enough to provide adequate cooling [27]. Moreover, the benefits of green and urban infrastructure are not evenly distributed across all communities, leading to social equity issues [27]. For example, there is concern about the uneven distribution and potential of green infrastructure increasing social inequality through eco-gentrification [38]. As cities invest in green and urban infrastructure, property values tend to increase, leading to gentrification and displacement of the very populations the infrastructure is meant to serve [38]. Additionally, there is a concern that green and urban infrastructure projects may be designed and implemented without adequate input or consultation from the homeless community, resulting in infrastructure that does not meet their needs [17]. Finally, there is a lack of funding for green and urban infrastructure projects, which can limit their effectiveness in mitigating the risks associated with extreme weather events for persons experiencing unsheltered homelessness [37].

### 1.6. Lack of Lived Experience Expertise 

Consistent with the pattern of social exclusion in homelessness, research also shows a lack of inclusion for PEH in homelessness services and policy planning [39]. This extends to municipal disaster planning initiatives, where the exclusion may lead to diminished adaptive capacity and make them disproportionately vulnerable to the negative impacts of extreme weather, leading to higher rates of mental illness, addiction, and interpersonal violence [40]. In a study [41] evaluating the inclusion of homeless experiences within climate action plans in California, only two out of fifteen plans formally recognized the vulnerability of homeless populations to climate impacts and successfully engaged with them through community outreach. However, those plans involved soliciting feedback through service providers rather than conversing directly with PEH. Minimal research has been conducted on how homeless individuals perceive and cope with severe weather conditions and how they attempt to overcome the resulting physical and physiological harm. This study examines the lived experiences of PEH and how they are affected by climate-change-induced severe weather. Specifically, we examine the vulnerabilities associated with extreme weather conditions and experiences of resilience and adaptation in coping with extreme weather events among the homeless community. 

## 2. Materials and Methods

### 2.1. Research Context 

Interviews were conducted at a local agency dedicated to assisting homeless individuals in the downtown area of Knoxville, Tennessee, a city within the Southeastern US. According to the *US 2020 Decennial Census Results* [42], the population of Knoxville is nearly 200,000 people, predominantly White (74.9%), with a nearly equal distribution of male and female residents. According to the same census results, approximately 20% of all residents live below the poverty line. Homelessness services are concentrated on one street, Broadway, which is located just north of the city’s downtown and largely devoid of tree canopy. A highway overpass spans one end of Broadway; PEH shelter under the overpass during weather events. The agency where recruitment was conducted is located on this street, and it provides day shelter, case management, and street outreach, as well as permanent supportive housing. The city’s homeless service providers utilize a “White Flag” policy to allow access to daytime indoor shelter during extreme temperatures, which are defined as less than 32 degrees Fahrenheit in the winter (<0 degrees Celsius) or above 90 degrees Fahrenheit in the summer (>32 degrees Celsius) [43].

### 2.2. Sample Recruitment and Participation 

The sample was recruited from PEH present at the agency during the recruitment dates—July 2022, July 2023, and September 2023. The dates corresponded with a research project that funded this study during summer months across these two years and the research assistants who were assisting in data collection (see funding acknowledgment). All recruitment and interviews were conducted in English. Forty-two individuals participated in the study, ranging in age from 24 to 65, with an average age of 49 (*SD* = 9.70). Nearly half (42.9%) identified as female, 50% identified as male, and 7.1% declined to answer. The majority (73.8%) identified as White, with 9.5% identifying as Black or African American. For most (61.5%), this was their first or second time experiencing homelessness. Nearly all participants had experienced extreme heat (88.1%), extreme cold (88.1%), and extreme rain or flooding (92.9%) while homeless. In addition, 40% of participants stated they had experienced some other form of extreme weather, such as hurricanes, severe thunderstorms, hail, snow, tornadoes, and extreme wind, while homeless and unsheltered. While we did not explicitly ask about whether persons were experiencing sheltered versus unsheltered homelessness, the agency where participants were recruited largely serves individuals who are sleeping outdoors, in a car, or staying in an emergency shelter. See Table 1 for complete demographics.

### 2.3. Ethical Protocol

The University of Tennessee’s IRB approved this study. Participants were required to be 18 years or older. Digital informed consent was required for participation. Participants were notified of their freedom to withdraw from this study at any point and abstain from answering questions during the demographic survey and semi-structured interviews. Participants gave their consent for interviews to be recorded and transcribed. All interviews were conducted in person, but Zoom was used to record them. Audio files were uploaded to a password-protected computer belonging to the PI, where they were subsequently anonymized and stored.

### 2.4. Data Collection 

Interviews for this study were structured to include a combination of cross-sectional, semi-structured, and in-depth (open-ended) questions. Participation was voluntary. Individuals currently experiencing homelessness and present at the agency during the interview timeframe were eligible to participate. Interviews lasted 20–45 min, and participants were offered a $10 gift card as compensation for their time. 

The study participants were requested to complete an online demographic survey as part of the research protocol. This was followed by six open-ended questions and follow-up probes regarding their experience with extreme weather while experiencing homelessness. The primary questions included the following: (1) *Since losing housing, tell me about a time when you’ve experienced what we would call “extreme weather”, like if it has been very hot or very cold or very rainy*. (2) *Has the impact of weather been different for you while homeless compared to when you had housing?* (3) *Have you had problems because of weather?*; (4) *What do you do to deal with extreme weather?* (5) *Are there places you wish you could access during extreme weather events?* (6) *Are there reasons you cannot access these places?* Questions were developed by two team members based on the primary research question—how do PEH experience extreme weather—and the theory of adaptive capacity—how do PEH try to reduce their risks during extreme weather events.

Interviews were conducted by four research team members according to a standardized protocol and checklist that was developed by the first author. She also trained the other team members and debriefed them to ensure adequate ability to develop a rapport with interviewees, ensure ethical compliance, and standardize interviewing procedures. Given the timing of the interviews during warmer weather, recall bias about cold weather might have influenced the result. Research team members were trained to mitigate this risk by probing about weather events in all seasons, including cold weather and rainy periods. While the questions asked about extreme weather, in general, most of the participants focused on extreme temperatures, and so the subsequent analyses focused on the experiences of extreme temperatures.

### 2.5. Data Analysis 

Data were transcribed using rev.com, a professional transcription service that uses human transcribers. Files were transcribed verbatim by paid transcribers who listened to the digital audio files of interviews. Through grounded theory methodology, we utilized axial and iterative coding methods to extract themes from the data through direct content analysis [44]. Software used in the process was limited to Microsoft Word (https://www.microsoft.com/ja-jp/microsoft-365/word) (accessed on 15 May 2024), Excel (https://www.microsoft.com/ja-jp/microsoft-365/excel) (accessed on 15 May 2024), and PowerPoint (https://www.microsoft.com/en-us/microsoft-365/powerpoint) (accessed on 15 May 2024). We systematically categorized data according to the emerging relationships, patterns, and themes during the initial coding process and iteratively revisited and refined codes throughout the analysis. This allowed us to identify patterns and relationships within the data and provide a comprehensive analysis. 

Two research team members initially coded the data and began to identify emergent subthemes separately. Two additional research team members then reviewed the initial subthemes to validate them and identify discrepancies. Once subthemes were validated, the team worked to group subthemes into larger themes, which were iteratively refined and validated through group discussion and validation against the raw data. Finally, the results were shared with the community partner for external validation. Given that each person’s experience is idiosyncratic, we chose not to quantify the themes. While individuals may describe an experience embodied in a given theme, their experience is slightly different.

## 3. Results

Results are organized according to the five key themes that emerged from the data analysis process described above. In short, the themes are as follows: (1) Adapting Is Impossible, (2) White Flag Policies, (3) Conflict and Density, (4) Community, and (5) No Place to Go…but the Bus. The first theme, Adaptation is Impossible, points to the reality that without shelter, true relief and shelter from the weather is impossible. The second theme, White Flag Policies, speaks to how agencies create policies that control and restrict PEH access to resources during extreme weather. The third theme, Conflict and Density, highlights how PEH are often crowded in the few areas that do provide some shelter during extreme weather and how this can create new social problems. The fourth theme, Community, is the opposite of the third theme and explains how reliance on others is a key strategy for building adaptive capacity among PEH. Finally, the fifth theme, No Place to Go, emerged due to the frequency with which participants cited the public bus as the one welcoming place where they could get relief from the weather.

### 3.1. Adapting Is Impossible 

Those experiencing unsheltered homelessness shared how the physical and psychological effects of severe weather are difficult to endure and almost impossible to escape. Participants with chronic health conditions discussed how extreme weather increased their physical pain and created difficulties with their individual capacity to cope or adapt. Other participants discussed how various attempts at adaptive methods failed and often placed them at greater risk of harm. For example, many participants discussed their measures to stay warm during cold weather events and the risks and harm they endured.


*“I camped [at the creek] for a long time. The second winter I was here. [...] I had hand sanitizer. You light it. Put it in a pan and light it, to burn it. Alcohol, it’s dangerous. It’s 91 percent. It caught me and the tent on fire.”*



*“We got stuck out past curfew at [Agency X], and we just wound up on one of the front porches of the [Agency Y] across the street. Just huddled against the cold. It got to the point where we were shivering so bad that we decided [...] the most sensible thing to do at that moment was to go find some methamphetamine to get the blood flowing a little bit better so we could walk around for the rest of the night without freezing to death.”*



*“Both my hips are metal and stuff, and so the coldness brings a lot of pain. And sometimes, when it freezes up, it wants to dislocate. So, I’ve had to go to the hospital, I think, about three or four times with my hip being put back in. [...] And during the summer, the muscles stick, and it pops out. So yeah, there’s been times where I felt like if they had took us in, it wouldn’t happen.”*



*“Yeah. When we had that cold snap and everything. Oh man, I was out there. No blanket or nothing. Trying to cover up with whatever I could get.”*



*“The whole winter long and we’d wake up to six inches of snow on your hood, and windows and stuff. We made it through with candles in the car [...] and had a little lantern.”*



*“I lay it [the poncho] on the ground now, and then I got a sheet, I put the sheet over top of that, and I got a little bitty baby blanket for a child or maybe a rug I think it is, but keeps you warm, keeps you alive.”*



*“I got arthritis. [...] The weather affects it or something is what they said. [...] It takes 20 min to warm up.”*


For some, attempting to survive extreme temperatures actually puts them at new risks.


*“The most sensible thing to do at that moment was to go find some methamphetamine to get the blood flowing a little bit better, so we could walk around for the rest of the night without freezing to death.”*


During times of extreme heat, participants shared how they found it extremely difficult to find safe spaces to sleep, stay out of the sun, and stay hydrated. Participants experiencing disabilities and a lack of mobility were further restricted and found few or no options to escape the heat and sun. For some participants, this led to heat-related injuries (e.g., dehydration, heat cramps).


*“There was nowhere else to go. I can’t walk far. You’re just stuck in the space where you are, and if there’s no shade, there’s no shade.”*



*“That’s one of the biggest issues that we have out on the streets is that we don’t get enough water or get enough fluids. Dehydration becomes an issue.”*



*“Oh, gosh. Well, I had passed out walking down the highway and didn’t have anything, no money. I was going to see my daughter and just left my husband. And I was so thirsty I picked up a bottle of water off the side of the road. Didn’t know who had been drinking it or anything. I smelled it and drank it because I was so thirsty. But that was like the worst day of my life, you know what I mean? That was really hard. It was so hot.”*



*“Cramps in the heat, yeah. At night when I’m sitting still and stuff, though, it just draws me up in knots. It’s from being dehydrated, but I drink water constantly. I try to drink Gatorade, but I can’t always afford it. So, yeah. Physically, cramps.”*


### 3.2. White Flag Policies

Agencies play a dual role for PEH, depending on the person and the agency involved. For some participants, an agency’s humane and flexible policies, such as “White Flag” policies, were critical to survival in extreme weather. However, other agency’s policies, such as banning illicit substances or requiring a medical necessity to stay indoors, exacerbated climate-related risks for unsheltered homeless persons and prevented shelter during times of extreme weather events. 


*“A lot of people do make it into the [Agency X], like get warm and stuff, but a lot of people still, for whatever reason, get stuck out on the street, whether they’re too high on drugs to come in or whether they’re just criminally trespassed or suspended. That happens to a lot of people around here because of behavioral issues or just getting caught with drugs in your bag or something like that on entry to the mission and other various reasons that people get criminally trespassed or suspended temporarily, resulting in them being stuck out on the street during harsh weather conditions, like the extreme cold and stuff. I’m sure there were probably a few deaths that probably happened because they were trespassed or something of that nature and couldn’t come into a shelter during that time. It’s unfortunate that there’s not really an alternative for some of those extreme situations.”*



*“When [people] get CT’ed [criminal trespassing citation], I heard they don’t get fed. [...] I’ve seen them be in wheelchairs and stuff getting CT’ed. And I’m thinking, that would be mostly people that needs to be in there...and they stopped feeding them...if you’re outside, you don’t get fed, and I think that’s crazy because it’s a homeless shelter.*



*“[Agency X] keep you outside unless they deemed it necessary for you to come inside. They made you go to a doctor and made you get a note saying that you had to be inside for heart problems, whatever, whether it was hot, cold. So, if there was any place [during the day] to be on the inside, it’d be nice.”*



*“Miserable. You about freeze to death. Churches sometimes come down there and hand out blankets and stuff, so that helps. I don’t stay inside [Agency X] because I used to be in a state’s custody when I was younger and we were in, I’m not sure what it was called, but there was beds just like those in there. I can’t stay up there.”*



*“Here you come in and get you some water anytime you want and they don’t turn you down.”*



*“Jackets and stuff like that? [Agency X] always does a thing in December, which they do the coats. They give people, they give homeless people an opportunity to get them at least one coat.”*



*“It was around Christmas time, I think, is when it got real, real cold those two weeks, and they had put the little heat tent by the [Agency X].”*


### 3.3. Conflict and Density

Dense living environments, driven in part by a paucity of homeless-friendly places, can produce social conflict and unsafe conditions. Participants discussed how they are often faced with the decision to endure the elements or go under the bridge (overpass) for shelter where there is potential for assault, drug and alcohol use, and being witness to violence and trauma.


*“We call [under the overpass] the Devil’s Playpen. [...] People do drugs, people overdose all the time down there and they drink down there. There’s always the possibility of a really bad vibe.”*



*“There’s always [under the] bridge, you’re out of the sun. But then there are also drugs, alcohol and fights, stabbing. You got to make choice. Are you going to bake? Or are you going to go under that bridge? And I chose to go under this bridge. I have experienced a lot of things from others under the bridge. I’ve been lucky nobody’s bothered me. But I didn’t really want to witness what was going on around me.”*



*“And then finding a place you can sit is another. They have places underneath the bridge, but it’s so crazy under there that I don’t even like going down there.”*


Some participants also discussed how restricted mobility and access to resources led to high concentrations of people in resource-rich areas. Specific to the geographic location of this study, homelessness services are concentrated within a single area downtown, leading to greater potential for conflict, as well as avoidance of resource-rich areas for some.


*“When it’s time to come in, you line up. You sit in line. You’re always in line for something. You’re in line to eat, and then after you eat, you go smoke, then there’s a line for the shower. [...] Then you’re in line for a bed, and then you’re in line for sheets, blankets, pillows. It’s all about lines and waiting. I would rather look up to the stars on a piece of cardboard and have peace of mind.”*



*“I do, yes, but I have an attitude problem, okay, with people saying crazy stuff to me. I got a mouth, so I choose to stay away from stuff like that [the bridge underpass] as much as possible because I know how my attitude can get towards people at times.”*


### 3.4. Community

While some participants discussed the issues and risks associated with being in high-density areas, others discussed how social ties helped them cope with severe weather. Participants described the value of having a friend to pass the time with and how they would help one another stay warm. For others, forming relationships with employees at local businesses mitigated the negative effects of unsheltered homelessness. Participants occasionally shared that they appreciated the resources provided to them by volunteers and ministries.


*“When it’s just me with nobody else to produce body heat and stuff like that, it’s hard. It can get real hard in the cold. [...] If I have a friend with me, we’ll just talk to get by the time, just talk about stuff and just talk about how we can get up out of it, about our way out of doing things differently.”*



*“[Volunteers] provide pretty much everything we need. They do. They give us anything we need like that. They’re good at providing food and soap and lotions. We get all that. You can get all that. You just have to learn where the places are [...] through word of mouth.”*



*“I’d go in there [fast food business] and they’d help me. They’d bring me food. The women that worked there in the morning, they would even make me food and bring it. They’d bring me coffee. That was very nice.”*



*“I did a training for fire safety, so they hand out these little blankets, they’re aluminum foil or something.”*


### 3.5. No Place to Go…but the Bus

Participants discussed how they experienced restricted mobility through the over-policing of parks and areas where homeless individuals may congregate, constant surveillance, and the stigmatization of the homeless community. Participants discussed being routinely asked to leave certain sidewalks or parks, preventing them from accessing shade in the summer or from standing in sunny areas during the winter. Others discussed the difficulty in finding a place to access safe drinking water or a clean restroom. Still, others discussed how they wanted a safe place to sit and stay out of the elements and rest.


*“It’s hard to find a place to get in. It’s hard to find a place to use the bathroom. It’s hard to find a place to get a drink of water.”*



*“When it’s cold, of course you’re looking for somewhere the sun’s shining to stand. But you’re not allowed to stand on sidewalks on Broadway.”*



*“I wish they had places that I could sit and get away from the sun.”*



*“There was nowhere else to go. I can’t walk far. You’re just stuck in the space where you are, and if there’s no shade, there’s no shade.”*


Participants specifically discussed how over-policing and stigmatization led to avoidance and exclusion from systems that would otherwise provide refuge. 


*“I had flooding out there. It was bad. It was. There wasn’t nowhere to go. And they’ve been running us off everywhere. If you’re staying out, if you’re not in a building, because I was in and out of places too. [...] I woke up and I was almost surrounded by water. Up and then sucked the tent down, because I slept up on the hill down here.”*



*“I mean, you guys have a home to go to, we don’t. If you’re on the ban list down here at [Agency X], you can’t get in there. If you go down here to the foyer, you get to be a member to get in there, then you get kicked out. I mean, I don’t know. It’s like they put you on all these ban lists and then when the nitty gritty comes, you have nothing else left but to go to the street.”*



*“When I first got down here, they’d let us work and I’d come around and do a lot of work for them throughout the day, just to be able to sleep out in the back or something. But everybody straightened up.”*



*“You have businesses that don’t won’t to allow you to be in there to stay cool.”*



*“The police will come by still. Okay. Yep. They’ll move you along like a herd of cattle.”*



*“They took out the picnic tables where we were coming down there.”*


The one public space that participants mentioned consistently as being inviting was public transit. Multiple participants described riding the bus as a way to get relief from the heat.


*“I just get on the bus. [...] Use the…bus for air conditioning.”*



*“But other than that. Well, one good thing, [the public transit], if it’s a hundred degrees or more, they’ll let you ride the bus free. But then people can’t just sit on the bus and ride all day.”*



*“You can ride the bus around for a certain amount of time, and then you have to get off that bus and switch to another bus. Some of the drivers, though, they don’t care as long as you’re being quiet and you don’t cause no ruckus or make a mess.”*



*“Well, they got air conditioner on it [the bus]. Yes, and I love it. Sometimes I’ll ride the bus for hours and hours.”*


## 4. Discussion

Our study represents a rare example of an empirical effort to more directly recognize and include PEH and their lived experiences in climate change research and to understand how the built environment of the urban landscape contributes to their adaptive capacity. As others have argued, talking directly to PEH about their experiences may help produce more inclusive and responsive policy planning and service designs [39,40]. Overall, our findings shed light on the lived experiences of PEH during extreme weather, and they contribute a new perspective on how social exclusion within the built environment compromises their adaptive capacity in the context of climate-induced weather risks. Consistent with other research, we found that persons experiencing temperature extremes and severe weather during homelessness were particularly vulnerable to the associated health and psychological risks [17]. Specifically, we found that homeless individuals within this geographical area experienced heat exhaustion, heat stroke, extreme sunburn, and hypothermia. Our data also revealed how exposure to extreme temperatures can have a profound impact on the health and well-being of individuals experiencing homelessness and that unsheltered homeless are particularly vulnerable to these effects with increased maladaptive coping. For example, many participants discussed the loss of limited resources due to extreme weather events and the inability to find or utilize refuge during these events. This led some participants to engage in risky behaviors like drug use or to seek refuge in places known to be dangerous. 

The findings of this research go beyond prior research, though, to highlight how social exclusion, which is reinforced by the urban environment, can reduce adaptive capacity and increase negative health impacts caused by extreme weather for PEH. Systematic exclusion from social institutions and economic systems for PEH leads to less recognition of their needs and risks during extreme temperatures and weather. Moreover, urban planning and surveillance appear to reinforce social exclusion by creating infrastructure that is less inclusive and demarcating seemingly public spaces that have controlled access. As a result, enduring extreme temperatures while unsheltered was physically and psychologically untenable. Attempts to cope and adapt during very cold nights or very hot days were often unsuccessful, leading to increased harm and possible death within the homeless community.

We found that service providers and local law enforcement within the urban environment play a critical role in helping or hindering people’s ability to survive during temperature extremes, as they control access to resources and space, both indoors and outdoors. However, for individuals experiencing addiction or suffering from mental health disorders, agency policies could impede access to shelter and exacerbate climate-induced extreme weather risks. Participants discussed how a lack of storage limited their ability to maintain the resources needed to cope with and endure extreme temperatures and that in seeking shelter, they would be required to give up resources they had worked hard to obtain, like backpacks and tents. Because of the requirements and probable loss of property, many homeless individuals opted to remain unsheltered during extreme weather events. Due to urban planning and the centralized location of resources within this area, we found mobility is restricted within the homeless community. This has led to greater conflict within the homeless community and resource avoidance for some homeless individuals. Yet, we also found that the social ties and community engagement developed within this centralized location were protective factors in coping with extreme weather for some participants. For example, participants discussed receiving resources from local community groups who are aware that the homeless community congregates in this specific area. Lastly, we found that stigmatization of the homeless community, over-policing of specific areas, and surveillance led to avoidance and exclusion from systems of refuge and an increased risk of physical and mental harm. 

## 5. Conclusions

Our participants’ rich descriptions of challenges and resources in the face of extreme weather offer insight into potential policy changes based on their lived experiences. Firstly, participants’ experiences underscore that shelter and housing are key to mitigating the risks associated with extreme weather events and that without basic access to shelter, their adaptive capacity is severely limited. This link between housing and adaptive capacity is consistent with a model of Housing First, which has been the widely accepted best practice for homelessness services within the US for nearly twenty years [45]. Continued efforts to ensure equity in safe and affordable housing is both a long-term solution to housing and environmental justice efforts, though. In the short term, mitigating the effects of extreme weather on the homeless community could be achieved through alternations to the urban infrastructure and emergency shelter to make them more inclusive and responsive to the changing climate. It is important to note that to be accessible, homeless-friendly infrastructure cannot just physically shield persons from environmental risk. It must also preserve privacy, mobility, and dignity. 

Our participants’ descriptions of how they navigate through the urban environment to find relief suggest, however, that urban infrastructure could increase adaptive capacity for PEH and other socially excluded groups if urban planners aimed for holistic and inclusive green designs. Among participants’ descriptions of their experiences, lack of water, dehydration, and heat stress symptoms were common. Such ailments suggest that immediate responses could include making water more available and accessible to the homeless community [46]. Warm-weather climate mitigation efforts should also include increasing shade by planting trees and making shady areas available to the homeless community, specifically by placing greenery and parks in homeless service sectors. In addition, attention should be given to preventing eco-gentrification of urban parks and green spaces, given that our participants spoke about how such areas that are currently in existence frequently are often not welcome to PEH. To mitigate the effects of cold weather, mixed-use daytime sheltering modeled after indoor streetscapes should be considered. 

Furthermore, our findings indicate that any effort to increase adaptive capacity during extreme weather must involve more direct healthcare services for PEH particularly. Given the social exclusion and barriers it creates to resources, providing mobile healthcare resources may be critical to reaching people. Examples might include on-the-ground clinics for in-person care, as well as more innovative solutions such as using drones to deliver bottled water and blankets, and other weather-mitigating resources to individuals sleeping in hidden or hard-to-reach locations.

Ironically, though, our findings also highlight how a culture of socially excluding PEH could create potential problems for urban planners as they try to design more inclusive green infrastructure. As our participants noted, there was strong resistance and efforts to police them out of the most desirable urban green spaces. In general, parks are intended for rest and recreation and not as shelter in lieu of building permanent housing. Thus, any urban design team ought to work in partnership with homeless service providers, as well as those currently or formerly homeless, to build and implement resources for adaptive capacity that are both responsive to their needs and feasible to implement. Beyond the physical engineering of public spaces, municipalities can adopt stronger community policing and trauma-informed street outreach within urban settings. Such training could help to shift the culture within law enforcement from surveillance to relationship building and support. In addition, municipalities can increase funding for interdisciplinary street outreach teams, which include law enforcement officers, as well as social workers, psychiatrists, psychologists, occupational therapists, doctors, and nurse practitioners, all working collaboratively to address the needs of PEH. During extreme weather events, it appears that such a culture shift could be highly beneficial to helping PEH seek necessary relief in public spaces rather than being pushed out of them. The goal should be to understand why and how PEH are utilizing public spaces and then to help them access the resources that they need to stay safe during extreme weather.

This research also calls attention to the need to rethink underpass and bridge spaces as areas used by the homeless community. There should be efforts made to increase the safety of these areas for all individuals. Installing lockers in accessible and safe areas could help PEH, given that many of our participants discussed difficulties storing and maintaining the supplies and gear necessary for enduring extreme weather events. Lastly, consideration should be given to trauma-informed training for local police departments and case managers working with homeless individuals to prevent further trauma and stigmatization of this vulnerable group. 

Finally, consistent with the theory of social exclusion, our results indicate that PEH often do not feel included in sustainability planning. Community planners, leaders, and municipal engineers should explicitly consider creating and developing plans and policies for extreme weather events that are inclusive of and responsive to PEH [2,18,39]. As an example, the needs of the homeless community should be included in adaptation and disaster planning [47,48]. Beyond nominal inclusion, plans and policies should be codefined with PEH, as well as those who work with them [39,48]. Moreover, research efforts such as this study should continue to be made to explore the individual and structural responses that currently enable safety and resilience within the homeless community. 

### 5.1. Limitations and Future Research

In this study, participants were recruited from a service-engaged sample within an agency setting, but future research should seek to recruit PEH at encampments, as many of these individuals do not frequent local agencies. Our findings are also limited by the timing of data collection during warmer weather months. Future research conducted during colder months may produce different risks and adaptive challenges based on different weather conditions. Likewise, studies that replicate this work in other geographic areas with different climatic conditions may also produce alternative results, for example, in places with near constant precipitation, such as the Pacific Northwest of the US, or areas with extremely high temperatures, such as the US Southwest.

Research extending from this study should also ask PEH directly what they think communities can do in terms of policy planning to better support them in the context of climate change. The knowledge collected during these interviews ultimately mapped how PEH traversed the city in response to temperatures, i.e., spaces were sought out, avoided, or deemed off-limits. Places assumed new meanings depending on the quality of refuge they offered. Future research to create a psycho-geographic map of the city based on the participants’ accounts could pinpoint where and how to design green infrastructure, e.g., tree canopy, water access, privacy, and safe spaces. Some participants noted that safety from extreme weather led to a lack of safety from others—choosing to seek refuge in places known to be dangerous, such as overpasses. Finding ways to design shelters for PEH may not only protect them from physical health risks during extreme weather but also interpersonal risks. Our areas of inquiry stemming from this study include exploring how public transit serves as a resource to build adaptive capacity within extreme temperatures, how this could be expanded, and the correlation between severe weather and interpersonal violence.

Finally, our findings highlight that municipal efforts to build more inclusive public infrastructure would likely face strong resistance from a general culture of social exclusion, as well as real public health concerns that stem from using public spaces for personal shelter. Urban planners seeking an inclusive green approach must recognize that public parks and water fountains are not designed as places of shelter or bathing. Still, efforts to criminalize homelessness or push people out of public spaces will only engender greater long-term social problems. As communities face more rising temperatures and a greater need for shelter among their residents, future research can help to highlight the need for housing as a climate change issue. Ultimately, housing is the greatest resource in building adaptive capacity against extreme temperatures; increasing and diversifying housing stock is essential for inclusive green infrastructure. 

### 5.2. Conclusions

The findings in this paper ultimately indicate that PEH need access to shelter as the strongest way to build their adaptive capacity in the context of extreme temperatures. Building new housing is a longer-term solution in most communities, though, and thus, for PEH, who disproportionately reside in urban environments, urban infrastructure can contribute to their risk or protection in the context of climate change. Green infrastructure can help increase their adaptive capacity, but only when the physical designs are paired with a culture of equity and social inclusion that promotes access and use for all members of the community. Distributing water throughout cities, increasing tree canopy and prioritizing underserved neighborhoods, offering daytime shelters, and training law enforcement and service providers to utilize trauma-informed and compassionate relationship-building skills are all strategies to help reduce social exclusion and promote adaptive capacity for PEH in the face of a changing climate.

## Figures and Tables

**Table 1 ijerph-21-00984-t001:** Sociodemographic characteristics of participants.

Sample Characteristics	*n*	%
Gender		
Female	18	42.9
Male	21	50.0
Prefer not to say	3	7.1
Race/Ethnicity		
White	31	73.8
Black or African American	4	9.5
American Indian/Native	2	4.8
Hispanic/Latino/a/x	1	2.4
Other	1	2.4
Prefer not to say	3	7.1
Highest Level of Education		
Some high school	14	33.3
High school degree	11	26.2
Post-high school training	3	7.1
Highest Level of Education		
Associate degree	3	7.1
Some college	7	16.7
Other	1	2.4
Current Employment		
Employed part-time	1	2.4
Unemployed, not looking	26	61.9
Unemployed, looking	11	26.2
Prefer not to answer	1	2.4
Marital Status		
Married	6	14.3
In a relationship, not married	2	4.8
Widowed	1	2.4
Single, divorced	21	50.0
Single, never married	8	19.0
Other	1	2.4
Prefer not to say	3	7.1
Mean Age	49.51 (*SD* = 9.69)	

## Data Availability

Data inquiries can be sent to the first author, Courtney Cronley (ccronle1@utk.edu).

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
