# Peer review of "Persons Experiencing Homelessness during Extreme Temperatures: Lessons for Promoting Socially Inclusive Adaptive Capacity"

_ijerph, 2024, doi:10.3390/ijerph21080984_

Round 1

Reviewer 1 Report

Comments and Suggestions for Authors

The study addresses a current and relevant issue. However, there is a major concern about the current status of the paper, specifically regarding it’s structure, flow and citations. The introduction has themes, that is unclear why they were selected as such. Further, there are different themes in the results as part of the analysis- which I have addressed specific comments below-. The discussion is short and does not tie the introduction themes with results themes. I recommend shortening the introduction. As well I recommend integrating themes from introduction and results into the discussion.

Future research section should be described before conclusions.

Limitations section of this study is missing.

Reorganize current conclusion. Identify discussion points and incorporate accordingly.  The conclusion should be conclusion statements, not to include references and further discussion. 

This paper is not ready to be published, needs major areas of review, careful reading and edits in every section. 

1.  It is advised to have keywords that are not included in the paper title. Consider a shorter title with less key words in it.

2. Provide a clean version of the paper having checked for spelling and not in “track changes”. 

3. Abstract could improve clarity by explicitly describing methods, results, discussion. However, provides good overview of the study purpose and some findings

4. Line 21. Check spelling “wlel”.

5. Lines 18-22. Condense this long sentence. 

6. Lines 35-44 could be reorganized to provide a better and concise flow. Include references. However, provide good insight to trends in homelessness. 

7. Line 45-52  missing relevant references, most seems to come from the same reference. 

8. PEH should be added to line 67 and 69 instead of “individuals experiencing homelessness” 

9. Line 71 need more references.

10.   Lines 78-81 repetitive, could be condensed to state something like: “Environmental conditions can influence access to necessities such as food, water, and shelter. Not having access to these can profoundly affect the mental health and well-being of adults and children, and particularly that of PEH [2,9]. 

11.  Line 93. Remove “Furthermore”. 

12.  There is a tendency in the writing style to include “connectors” such as Furthermore, sometimes repetitive, and unnecessary. 

13.  Lines 100-102. Make sure to cite appropriately. This reference does not explicitly talk about heat related illnesses. 

14.  Line 104. Consider “… hypothermia, respiratory, and cardiovascular…”.

15.  Line 106-107. Again, verify your track changes, punctuation marks and correct references. 

16.  Add PEH to line 140, make sure consistent throughout paper

17.  Line 145-147. Correct and cite appropriately.

18.  Add PEH to line 157

19.  Add PEH to line 158. Again, be consistent throughout the paper.

20.  Lines 158-161 could be condensed; green and urban infrastructure could be abbreviated to GUI if used throughout paper. However, also think this may not be appropriate here. Sounds like hypothesis or purpose statement I’d expect at end of introduction.

21.   Lines 166-171 lacking references and clarity.

22.  Lines 172-175 lacking references

23.   Again, would remove the transition “though” from line 178, too informal.

24.  Note: Throughout the document, there is a tendency to write two contiguous sentences citing the same reference, please correct accordingly. Section 1.6 has a couple of these examples in each paragraph.  

25.  Section 1.6. Merge to 1 paragraph. 

26.  Would adjust sentence beginning in line 191 to state” However, those plans involved soliciting feedback through service providers rather than conversing directly with PEH”; and would move reference number to the end of that sentence and remove from the two previous sentences.

27.  Also, reference numbers 31 and 32 appear before reference number 30, adjust accordingly.

28.  Correct and cite appropriately in lines 201-214

29.  Provide reference for white flag policy described in line 211 

30.  Elaborate/clarify recruitment dates in lines 216-217: were participants only recruited during those three months? In which languages? Elaborate why so distant in time for interviews to happen, and mention recall bias.

31.  Include a map to orient the audience on geographic location of the City in Southestern US, the city, intersection where interviews and or shelters happen, as this is an international journal

32.  Table 1 needs formatting, consider putting age at the top

33.  Section 2.4. Please described the interviewer(s), who trained them and/or experience in this topic, was always the same interviewer? 

a.        Are there references as of why these questions were selected? Who created these questions/why? Have they been proven elsewhere?

34.  Lines 291-292 repetitive from section 2.3

35.  Please describe transcription method used, mentioned in line 292

36.  Please describe if any software was used throughout the coding process, including but not limited to Excel or platforms like Dedoose and Nvivo

37.  Could be helpful to provide de-identified study ID’s with their respective quotes, for example: “xxx quote” -Subject 2, F

38.  Results section. Please elaborate on the previous paragraph How/why did you come up with these themes? Is not clear.

39.  It is mentioned that there were participants with chronic conditions, was there any question/comment regarding chronic medications refills or access?

40.  Quote re-used as example for line 341, fix and remove accordingly

41.  Lines 376-377. Provide reference of mentioned agencies.

42.  Line 398. Add pending quotation mark 

43.  Quote line 458 is missing context.

44.  Add PEH to line 554, again be consistent throughout paper

45.  Again, consistency. PEH to line 631

46.  References: Line 720: The doi link is incorrect, this is unacceptable. Review all references.

It is imperative to revise all your references and their adequate citation before submitting a manuscript. 

Comments on the Quality of English Language

Author Response

1. Summary

Thank you very much for taking the time to review this manuscript. Please find the detailed responses below and the corresponding revisions/corrections highlighted in red font in the re-submitted files.

2. Questions for General Evaluation

Reviewer’s Evaluation

Response and Revisions

Does the introduction provide sufficient background and include all relevant references?

Must be improved

All revisions are detailed below in response to the reviewer’s specific comments.

Is the research design appropriate?

Must be improved

Are the methods adequately described?

Must be improved

Are the results clearly presented?

Must be improved

Are the conclusions supported by the results?

Must be improved

3. Point-by-point response to Comments and Suggestions for Authors

Comment 1: The introduction has themes, that is unclear why they were selected as such.

Response 1: The introduction includes subheadings for different key concepts that the authors feel are necessary to review and which ground the paper.

Comment 2: Further, there are different themes in the results as part of the analysis- which I have addressed specific comments below.

Response 2: We have clarified in the results that the subheadings refer to the themes that resulted from the data analysis.

Comment 3: As well I recommend integrating themes from introduction and results into the discussion.

Response 3: Discussion has been revised accordingly.

Comment 4: Future research section should be described before conclusions.

Response 4: Future research is now discussed before the final conclusions.

Comment 5: Limitations section of this study is missing.

Response 5: Limitations is now a separate section.

Comment 6: Reorganize current conclusion. Identify discussion points and incorporate accordingly.  The conclusion should be conclusion statements, not to include references and further discussion. 

Response 6: Conclusion has been revised to eliminate references and future discussion and summarize key findings.

Comments 7: It is advised to have keywords that are not included in the paper title. Consider a shorter title with less key words in it.

Response 7: Thank you for this recommendation. We have revised and shortened the title.

Comment 8: Provide a clean version of the paper having checked for spelling and not in “track changes.”

Response 8: The manuscript has been revised and thoroughly checked for spelling. Changes made in response to reviewer comments have been in red text, but the rest of the paper has been cleared of any track changes.

Comments 9: Abstract could improve clarity by explicitly describing methods, results, discussion. However, provides good overview of the study purpose and some findings

Response 9: We have revised the abstract to include more details about the study’s methodology.

Comments 10: Line 21. Check spelling “wlel”.

Response 10: We were unable to locate this spelling error in the manuscript.  

Comments 11: Lines 18-22. Condense this long sentence. 

Response 11: This content of this sentence has been restructured.

Comments 12: Lines 35-44 could be reorganized to provide a better and concise flow. Include references. However, provide good insight to trends in homelessness. 

Response 12: This paragraph has been restructured and the citation repeated throughout. All of the data reported in this paragraph come from the same source – the 2023 Annual Homelessness Assessment Report to Congress.  

Comments 13: Line 45-52 missing relevant references, most seems to come from the same reference. 

Response 13: Relevant references are included in the text.

Comments 14: PEH should be added to line 67 and 69 instead of “individuals experiencing homelessness” 

Response 14: We appreciate this observation and have changed the language to PEH. We have done so throughout the paper to be consistent with our terminology.

Comments 15: Line 71 need more references.

Response 15: A reference is listed for Line 71.

Comments 16: Lines 78-81 repetitive, could be condensed to state something like: “Environmental conditions can influence access to necessities such as food, water, and shelter. Not having access to these can profoundly affect the mental health and well-being of adults and children, and particularly that of PEH [2,9]. 

Response 16: We appreciate this recommendation and have revised the sentence accordingly.

Comments 17: Line 93. Remove “Furthermore”. 

Response 17: “Furthermore” has been removed.  

Comments 18: There is a tendency in the writing style to include “connectors” such as Furthermore, sometimes repetitive, and unnecessary. 

Response 18: We have revised the manuscript with an eye to deleting unnecessary and repetitive connectors. Thank you for this suggestion.

Comments 19: Lines 100-102. Make sure to cite appropriately. This reference does not explicitly talk about heat related illnesses. 

Response 19: We have identified this error and replaced the citation with a publication that more accurately reflect our support our statement.  

Comments 20: Line 104. Consider “… hypothermia, respiratory, and cardiovascular…”.

Response 20: Thank you for this suggestion. The sentence has been revised.

Comments 21: Line 106-107. Again, verify your track changes, punctuation marks and correct references. 

Response 21: This section of text has been revised with an extraneous sentence deleted.  

Comments 22: Add PEH to line 140, make sure consistent throughout paper

Response 22: This line has been revised as recommended.  

Comments 23: Line 145-147. Correct and cite appropriately.

Response 23: The text has been corrected and citation verified.

Comments 24: Add PEH to line 157

Response 24: This has been done.

Comments 25: Add PEH to line 158. Again, be consistent throughout the paper.

Response 25: Again, done.

Comments 26: Lines 158-161 could be condensed; green and urban infrastructure could be abbreviated to GUI if used throughout paper. However, also think this may not be appropriate here. Sounds like hypothesis or purpose statement I’d expect at end of introduction.

Response 26: We appreciate the reviewer’s suggestion and have deleted this paragraph.  

Comments 27: Lines 166-171 lacking references and clarity.

Response 27: This section has been revised for clarity and additional references cited.  

Comments 28: Lines 172-175 lacking references

Response 28: References have been added.  

Comments 29: Again, would remove the transition “though” from line 178, too informal.

Response 29: “Though” has been removed.  

Comments 30: Note: Throughout the document, there is a tendency to write two contiguous sentences citing the same reference, please correct accordingly. Section 1.6 has a couple of these examples in each paragraph.  

Response 30: We have revised Section 1.6 to address this concern and have made additional revisions throughout the paper where the repeated use of a citation could be problematic.  

Comments 31: Section 1.6. Merge to 1 paragraph. 

Response 31: This section has been merged into a single paragraph.  

Comments 32: Would adjust sentence beginning in line 191 to state” However, those plans involved soliciting feedback through service providers rather than conversing directly with PEH”; and would move reference number to the end of that sentence and remove from the two previous sentences.

Response 32: The sentence has been revised as suggested.  

Comments 33: Also, reference numbers 31 and 32 appear before reference number 30, adjust accordingly.

Response 33: References have been revised.  

Comments 34: Correct and cite appropriately in lines 201-214

Response 34: Citation has been corrected.  

Comments 35: Provide reference for white flag policy described in line 211 

Response 35: A reference and in-text citation for the “White Flag” policy have been added.  

Comments 36: Elaborate/clarify recruitment dates in lines 216-217: were participants only recruited during those three months? In which languages? Elaborate why so distant in time for interviews to happen, and mention recall bias.

Response 36: We have added more details to sections 2.2 and 2.4 regrading participant recruitment and recall bias.

Comments 37: Include a map to orient the audience on geographic location of the City in Southestern US, the city, intersection where interviews and or shelters happen, as this is an international journal

Response 37: Due to the confidentiality of participants and the partner agency, we are unable to provide a map of the city and the specific intersection. However, in section 2.1 we do provide a description of the urban design and the location of the agency.

Comments 38: Table 1 needs formatting, consider putting age at the top

Response 38: Table 1 has been reformatted. We chose to leave age at the bottom but changed how we noted the statistics that we are reporting for age.  

Comments 39: Section 2.4. Please described the interviewer(s), who trained them and/or experience in this topic, was always the same interviewer? 

a.         Are there references as of why these questions were selected? Who created these questions/why? Have they been proven elsewhere?

Response 39: We have revised Section 2.4 to add more information about the training for and standardization of data collection procedures, as well as how the questions were derived.

Comments 40: Lines 291-292 repetitive from section 2.3

Response 40: Thank you for noting this erroneous repetition. We have deleted the second description of these procedures.

Comments 41: Please describe transcription method used, mentioned in line 292

Response 41: Additional information about the transcription method has been added to Section 2.5.  

Comments 42: Please describe if any software was used throughout the coding process, including but not limited to Excel or platforms like Dedoose and Nvivo

Response 42: We have added the information on what software was used in the data analysis process.

Comments 43: Could be helpful to provide de-identified study ID’s with their respective quotes, for example: “xxx quote” -Subject 2, F

Response 43: We appreciate this suggestion, but the IDs would not add any descriptive information to the text.

Comments 44: Results section. Please elaborate on the previous paragraph How/why did you come up with these themes? Is not clear.

Response 44: We have added an additional paragraph elaborating on the data analysis process and the results themes. (see Lines 311-316)

Comments 45: It is mentioned that there were participants with chronic conditions, was there any question/comment regarding chronic medications refills or access?

Response 45: Participants spoke to chronic health conditions such as asthma and clinical anxiety. The team did not directly ask about prescription drug access or refill, and participants did not comment on this.

Comments 46: Quote re-used as example for line 341, fix and remove accordingly

Response 46: The repetitive quote has been removed.  

Comments 47: Lines 376-377. Provide reference of mentioned agencies.

Response 47: We are unable to directly name agencies due to confidentiality.  

Comments 48: Line 398. Add pending quotation mark up with these themes? Is not clear.

Response 48: Quotation marks are provided.  

Comments 49: Quote line 458 is missing context.

Response 49: Context has been added.  

Comments 50: Add PEH to line 554, again be consistent throughout paper

Response 50: Original text has been replaced with PEH.

Comments 51: Again, consistency. PEH to line 631

Response 51: Likewise, PEH has replaced original text.  

Comments 52:  References: Line 720: The doi link is incorrect, this is unacceptable. Review all references.

Response 52: All doi links have been checked and any errors have been corrected. Thank you for noting this mistake.  

4. Response to Comments on the Quality of English Language

N/A

Reviewer 2 Report

Comments and Suggestions for Authors

The topic of this paper is important and timely, and gives needed voice to a marginalized population. The concrete suggestions for what could be done in the conclusion are clear and helpful. The following suggestions could improve the paper further:

-Line 20: The sentence that states “the lack of, and concurrent, policing of scarce resources“ is confusing—does this mean there is a lack of resources, while also concurrent policing of them? Please write this in a more clear way.    

-Line 21: please fix typo, should be “well” rather than “wlel”

-Line 26: please add period

-Line 42-43, 61, 62, 107, 158, 159, 167, 171, 195, 376, 377, 395, 396: please fix red/track change words

-Materials and Methods: Is there data on what percent of homeless individuals identified as unsheltered vs. sheltered homeless? Given that much of the intro focuses on this, it seems relevant to include.

-Line 281: This describes the “primary questions,” but were there other questions asked? If so, what were they?

-Results: please comment on whether any interviews were double coded and measured for inter-coder agreement and if not, why not.

-Is there a way to quantify what percent of respondents identified each theme? The themes are helpful, but would be even more so if it were clear what percent of participants identified each theme. If not, may be helpful to explain why this was not possible.

Author Response

1. Summary

Thank you very much for taking the time to review this manuscript. Please find the detailed responses below and the corresponding revisions/corrections highlighted in red text in the re-submitted files.

2. Questions for General Evaluation

Reviewer’s Evaluation

Response and Revisions

Does the introduction provide sufficient background and include all relevant references?

Yes

All revisions are detailed below in response to the reviewer’s specific comments.

Is the research design appropriate?

Yes

Are the methods adequately described?

Can be improved

Are the results clearly presented?

Yes

Are the conclusions supported by the results?

Yes

3. Point-by-point response to Comments and Suggestions for Authors

Comment 1: Line 20: The sentence that states “the lack of, and concurrent, policing of scarce resources“ is confusing—does this mean there is a lack of resources, while also concurrent policing of them? Please write this in a more clear way.  

Response 1: This sentence has been revised.  

Comment 2: Line 21: please fix typo, should be “well” rather than “wlel”

Response 2: Thank you for noting this error. We have corrected it in the text.

Comment 3: Line 26: please add period

Response 3: Period has been added.

Comment 4: Line 42-43, 61, 62, 107, 158, 159, 167, 171, 195, 376, 377, 395, 396: please fix red/track change words

Response 4: We apologize for the oversight and have fixed track changes.

Comment 5: Materials and Methods: Is there data on what percent of homeless individuals identified as unsheltered vs. sheltered homeless? Given that much of the intro focuses on this, it seems relevant to include.

Response 5: While participants were not explicitly asked if they were sheltered or not, the service provider where we recruited provided services largely to those unsheltered or staying in emergency shelter. We have noted this in the sample description.

Comment 6: Line 281: This describes the “primary questions,” but were there other questions asked? If so, what were they?

Response 6: Additional questions were limited to probing such as, “tell me more.” We have noted this in the text.  

Comment 7: Results: please comment on whether any interviews were double coded and measured for inter-coder agreement and if not, why not.

Response 7: We have provided a detailed description of the data analysis process, including double-coding.

Comment 8 :Is there a way to quantify what percent of respondents identified each theme? The themes are helpful, but would be even more so if it were clear what percent of participants identified each theme. If not, may be helpful to explain why this was not possible.

Response 8: We have explained in the text why we chose not to quantify themes. Given that each person’s experience is idiosyncratic, we chose to preserve the sense of this uniqueness in that each person, even if they have an experience embodied in the theme, experience it slightly differently.

4. Response to Comments on the Quality of English Language

N/A

Reviewer 3 Report

Comments and Suggestions for Authors

In this study, the authors explore effects of extreme temperatures on homeless populations in a Southeastern US city. The study is very well framed with a good literature review and establishes the need for this kind of research well. The focus on lived experience and storytelling through quotes gives much needed voice to this marginalized community. The discussion and conclusion summarize the findings and recommend relevant policy solutions clearly and in direct response to the findings. I have only minor questions for the authors to consider prior to publication, and very few suggestions offered in the detailed comments below.

As the focus of the study is on lived experience of those interviewed, did you gather evidence of policy solutions or practices from the study participants themselves? There are several that are implicit from the responses, but I am curious if this was asked directly or indirectly of the participants and whether that could be included as a section in the results. The policy recommendations in the discussion are relevant and reasonable based on what you have found in the study, but I do perceive a slight disconnect or shift when moving between these sections. Perhaps some additional language connecting the findings to the individual policies recommended would enhance flow through the article.

Additionally, while you do mention some relevant points in the further research section, are there other limitations you should note in this research? Perhaps considerations that are unique to this climate or the extent to which we can extrapolate to other climates? Or anything else unique to this population, the data collection methods, time of year of collection, etc. that may be relevant?

All in all, I believe this study is mostly ready for publication barring the few considerations described here and will make a useful contribution to the literature once published. I would recommend only minor revisions and would be happy to review again when ready. Thank you for the opportunity to review this manuscript, and for focusing on this important topic.

Detailed comments

General – This appears to be a tracked-changes version of the file. I still see some red underlined text.

L21 – “wlel” should be “well”

L45 – I might recommend introducing the PEH term earlier since so much attention is already placed on this population from the beginning of the article.

Table 1 – Perhaps a formatting issue, but I am not seeing the M or SD columns populated. Are these even necessary for the stats reported here? If only Age reports these statistics, you may consider removing that row from the table to report in the text only. The formatting for the second section on page 6 is also misaligned, with every other column shifted to the right. I recommend revising this table to correct these errors.

L344-346 – This quote is used in the section above as well. Do you mean for it to be repeated?

Author Response

1. Summary

Thank you very much for taking the time to review this manuscript. Please find the detailed responses below and the corresponding revisions/corrections highlighted in red font in the re-submitted files.

2. Questions for General Evaluation

Reviewer’s Evaluation

Response and Revisions

Does the introduction provide sufficient background and include all relevant references?

Must be improved

All revisions are detailed below in response to the reviewer’s specific comments.

Is the research design appropriate?

Must be improved

Are the methods adequately described?

Must be improved

Are the results clearly presented?

Must be improved

Are the conclusions supported by the results?

Must be improved

3. Point-by-point response to Comments and Suggestions for Authors

Comment 1: As the focus of the study is on lived experience of those interviewed, did you gather evidence of policy solutions or practices from the study participants themselves? There are several that are implicit from the responses, but I am curious if this was asked directly or indirectly of the participants and whether that could be included as a section in the results. The policy recommendations in the discussion are relevant and reasonable based on what you have found in the study, but I do perceive a slight disconnect or shift when moving between these sections. Perhaps some additional language connecting the findings to the individual policies recommended would enhance flow through the article.

Response 1: We did not directly ask participants about policy recommendations, but we agree with the reviewer that there are several implicit recommendations based on their descriptions of how they deal with extreme weather. We have revised the conclusion section of the manuscript to better address the shift from participants descriptions to implied policy recommendations.

Comment 2: Additionally, while you do mention some relevant points in the further research section, are there other limitations you should note in this research? Perhaps considerations that are unique to this climate or the extent to which we can extrapolate to other climates? Or anything else unique to this population, the data collection methods, time of year of collection, etc. that may be relevant?

Response 2: We agree with the reviewer and appreciate drawing our attention to this oversight. We have increased our discussion of limitations to the study and ways that future research can overcome these limitations.

Comment 3: L21 – “wlel” should be “well”

Response 3: Thank you for noting this error. We have corrected it in the text.

Comment 4: L45 – I might recommend introducing the PEH term earlier since so much attention is already placed on this population from the beginning of the article.

Response 4: Thank you for this suggestion. We have revised the introductory paragraph to use the “persons experiencing homelessness (PEH)” term.

Comment 5: Table 1 – Perhaps a formatting issue, but I am not seeing the M or SD columns populated. Are these even necessary for the stats reported here? If only Age reports these statistics, you may consider removing that row from the table to report in the text only. The formatting for the second section on page 6 is also misaligned, with every other column shifted to the right. I recommend revising this table to correct these errors.

Response 5: We have revised Table 1 accordingly.

Comment 6: L344-346 – This quote is used in the section above as well. Do you mean for it to be repeated?

Response 6: Thank you for noting this erroneous repetition. We have deleted the second use of the quote.

4. Response to Comments on the Quality of English Language

N/A